# Val66Met Polymorphism Is Associated with Altered Motor-Related Oscillatory Activity in Youth with Cerebral Palsy

**DOI:** 10.3390/brainsci12040435

**Published:** 2022-03-24

**Authors:** Michael P. Trevarrow, Hannah Bergwell, Jennifer Sanmann, Tony W. Wilson, Max J. Kurz

**Affiliations:** 1Institute for Human Neuroscience, Boys Town National Research Hospital, Omaha, NE 68010, USA; michael.trevarrow@boystown.org (M.P.T.); hannah.bergwell@boystown.org (H.B.); tony.wilson@boystown.org (T.W.W.); 2Department of Genetic Medicine, Munroe-Meyer Institute, University of Nebraska Medical Center, Omaha, NE 68198, USA; jsanmann@unmc.edu; 3Department of Pharmacology and Neuroscience, Creighton University, Omaha, NE 68178, USA

**Keywords:** genetics, BDNF, neurogenetics, MEG, beta, leg, frog game

## Abstract

Brain-derived neurotrophic factor (BDNF) plays a critical role in the capacity for neuroplastic change. A single nucleotide polymorphism of the *BDNF* gene is well known to alter the activity-dependent release of the protein and may impact the capacity for neuroplastic change. Numerous studies have shown altered sensorimotor beta event-related desynchronization (ERD) responses in youth with cerebral palsy (CP), which is thought to be directly related to motor planning. The objective of the current investigation was to use magnetoencephalography (MEG) to evaluate whether the *BDNF* genotype affects the strength of the sensorimotor beta ERD seen in youth with CP while youth with CP performed a leg isometric target matching task. In addition, we collected saliva samples and used polymerase chain reaction (PCR) amplification to determine the status of the amino acid fragment containing codon 66 of the *BDNF* gene. Our genotyping results identified that 25% of the youth with CP had a Val66Met or Met66Met polymorphism at codon 66 of the *BDNF* gene. Furthermore, we identified that the beta ERD was stronger in youth with CP who had the Val66Met or Met66Met polymorphism in comparison to those without the polymorphism (*p* = 0.042). Overall, these novel findings suggest that a polymorphism at the *BDNF* gene may alter sensorimotor cortical oscillations in youth with CP.

## 1. Introduction

Cerebral palsy (CP) results from an insult to the developing brain and is the most common pediatric motor disability [1,2,3]. Inherent to the disorder are mobility issues that stem from spasticity, hyperexcitable reflexes, sensory processing deficits, joint contractures, and deficits in motor planning [4,5,6,7,8,9,10,11,12,13,14,15]. Although physical therapy is the primary avenue used to optimize the motor actions of youth with CP, the therapeutic response is considerably variable, with some individuals experiencing vast sensorimotor advances and others being clear non-responders [16]. This response variability represents one of the primary unresolved complications in our understanding of the altered motor actions seen in this patient population. 

There has been a growing interest in determining whether there are underlying genetic factors that might influence the observed motor actions of youth with CP and an individual’s capacity for neuroplastic change [17]. Specifically, the protein brain-derived neurotrophic factor (BDNF) has been shown to play a prominent role in regulating the extent of the neuroplasticity. For example, outcomes from animal models have demonstrated that there is an up-regulation of BDNF within the sensorimotor cortices while learning novel motor skills and when adult motor patterns are shaped [18,19], and this ultimately can contribute to reorganization of the local sensorimotor neuronal networks representing the motor action [18]. Alternatively, inhibiting the activity-dependent release of BDNF results in hindered performance and decreased sensorimotor cortical reorganization [20]. A single nucleotide polymorphism at codon 66 of the *BDNF* gene that results in a valine to methionine amino acid substitution at one or both alleles (i.e., Val66Met or Met66Met) results in decreased activity-dependent release of BDNF and an adverse effect on neuroplastic change [21]. Transcranial magnetic stimulation (TMS) studies have also identified that neurotypical adults that have the *BDNF* genetic polymorphism exhibit a reduction in cortical excitability and limited cortical reorganization after practicing a motor skill [21,22,23]. Altogether these results illustrate that the *BDNF* polymorphism might impact the motor actions of some youth with CP.

Numerous studies utilizing magnetoencephalography (MEG) and electroencephalography have identified that there is a sharp decrease in the power of sensorimotor beta cortical oscillations that begins several hundreds of milliseconds before movement onset and lasts throughout movement duration [24,25,26,27,28,29,30]. This frequency-specific response, termed the peri-movement beta event-related desynchronization (ERD), has been extensively linked with motor planning and execution parameters [31,32,33,34,35,36]. Our laboratory has demonstrated that the sensorimotor beta ERD is stronger in youth with CP in comparison to neurotypical controls while performing a lower extremity target-matching task [36,37,38]. The current consensus is that the stronger ERD (i.e., greater power reduction) indicates that fewer pyramidal neurons in the local cortical area are oscillating at the beta frequency, while youth with CP plan and produce leg motor actions. This interpretation is aligned with the prior TMS literature that has identified individuals with CP as having less activation of the corticospinal pathways and an inability to modulate the excitability of motor pathways [39]. Potentially, these aberrant beta sensorimotor cortical oscillations might partially stem from the impact of the perinatal insult on the long-term optimization of neural generators that are involved in the production of motor actions. Conceptually, these cascading effects may have a greater impact on youth with CP who have a polymorphism at the *BDNF* gene. However, this conjecture has yet to be tested.

The objective of this investigation was to evaluate whether the *BDNF* genotype is predictive of the strength of the movement-related beta ERD response seen in individuals with CP. To this end, we used MEG to image the movement-related beta ERD while youth with CP performed a leg isometric target matching task and used polymerase chain reaction amplification to determine the status of the amino acid fragment containing codon 66 of the *BDNF* gene. Participants were subsequently categorized into groups that had a Val66Met or Met66Met *BDNF* genotype and those that had a Val66Val BDNF genotype, and the strength of their beta ERD responses were compared. We hypothesized that those with a polymorphism would have a more aberrant beta ERD than those without a polymorphism at the *BDNF* gene.

## 2. Materials and Methods

Twenty youth with spastic CP (age = 16.40 ± 4.37 yrs., females = 12) were recruited from local clinics and the community to participate in this investigation. The participants were without an orthopedic surgery or anti-spasticity treatments within the last 6 months, a dorsal rhizotomy, and/or clinical diagnosis of an arterial ischemic stroke or middle cerebral artery stroke. Individuals who had incurred a stroke were not included in this investigation because they are associated with large volume loss that would likely impact the cortical surface. The Institutional Review Board reviewed and approved the protocol for this investigation. All the parents provided written consent for their child to participate in the investigation and the youth assented.

Saliva samples were collected from the participants using the Oragene kit (DNA Genotek, Ottawa, ON, Canada). Genomic DNA was isolated according to standard laboratory procedures using a manual DNA extraction protocol and was quantified using the NanoDrop ND-1000^®^spectrophotometer (NanoDrop Technologies, Wilmington, DE, USA). Polymerase chain reaction (PCR) amplification of the 274-bp fragment containing codon 66 of the BDNF gene (RefSeq NM_170735.5) was performed according to standard procedures [40]. The PCR products were analyzed by direct sequence analysis in both the forward and reverse directions utilizing automated fluorescence dideoxy sequencing methods to determine the amino acid status at codon 66. Participants with the BDNF polymorphism included those with either the Val66Met or Met66Met genotype, while participants without the polymorphism had the Val66Val genotype.

The MEG methods described in the following sections are the same as what has been previously published by our laboratory when quantifying the sensorimotor cortical oscillations associated with the production of leg motor actions [36,37,38] The neuromagnetic responses were sampled continuously at 1 kHz with an acquisition bandwidth of 0.1–330 Hz using an MEGIN MEG system (Helsinki, Finland). All recordings were conducted in a one-layer magnetically shielded room with active shielding engaged for advanced environmental noise compensation. A custom-built head stabilization device that consisted of a series of inflatable airbags surrounded the head and filled the void between the head and MEG helmet. This system stabilized the head and reduced the probability of any large head movements during the data collections.

The youth were seated upright in a magnetically silent chair during the experiment. A custom-built magnetically silent force transducer was used to measure the isometric knee extension forces generated by the youth as they completed a Frog Task that we have used in our prior studies [41]. The Frog Task experimental paradigm involved the participant generating an isometric knee extension force with their right leg, between 15 and 30% of their maximum isometric force, to move a frog vertically and match a target, which appeared as a moth (Figure 1). The participants were instructed to match the presented targets as quickly and as accurately as possible. A successful match occurred when the moth that represented the target force was inside of the frog’s mouth for 300 ms, and the amount of force required by the participant was presented randomly. The stimuli were shown on a back-projection screen that was approximately 1 m in front of the participant at eye-level. Each trial was 10,000 ms in length. The participants started each trial at rest, fixating on the center of the screen for 5000 ms. After this rest period, the target moth appeared, prompting the participant to generate the isometric knee extension and match the target. The target was available to be matched for up to 5000 ms. Once either the target was matched or 5000 ms elapsed, feedback was given to indicate the end of the trial, and the participant returned to rest and fixated on the center of the screen while waiting for the next target to appear. The participants completed 120 target matching trials.

For the MEG experiment, four coils were attached to the head of the participant and used for continuous head localization throughout the experiment. The location of these coils, three fiducial points, and the scalp surface were digitized to determine their three-dimensional coordinates (Fastrak 3SF0002, Polhemus Navigator Sciences, Colchester, VT, USA). An electric current with a unique frequency label (e.g., 322 Hz) was fed to each of the four coils during the MEG experiment. This induced a measurable magnetic field, which allowed for each coil to be localized in reference to the sensors throughout the recording session. Since the coil locations were also known in head coordinates, all MEG measurements could be transformed into a common coordinate system. With this coordinate system (including the scalp surface points), each participant’s MEG data were co-registered with native space neuroanatomical MRI data using the three external landmarks (i.e., fiducials) and the digitized scalp surface points prior to source space analyses. The neuroanatomical MRI data were aligned parallel to the anterior and posterior commissures and transformed into standardized space. The MRI data were acquired with a 3T Siemens Skyra scanner (Munich, Germany) using a 32-channel head coil (TR: 2400 ms; TE: 1.94 ms; field of view: 256 mm; slice thickness: 1 mm with no gap; in-plane resolution: 1.0 × 1.0 mm). 

Using the MaxFilter software (MEGIN; Helsinki, Finland), each MEG dataset was individually corrected for head motion and subjected to noise reduction using the signal space separation method with a temporal extension [42]. Artifact rejection was performed at the level of the entire trial and was based on a fixed threshold method, supplemented with visual inspection. The continuous magnetic time series was divided into epochs of 4500 ms in length, from −2000 to 2500 ms, with 0 ms defined as movement onset. The baseline was defined as −2000 to −1500 ms. 

Artifact-free epochs for each sensor were transformed into the time-frequency domain using complex demodulation and averaged over trials [43]. These sensor-level data were normalized per time-frequency bin using the respective bin’s mean power during the baseline. The specific time-frequency windows used for imaging were determined by statistical analysis of the sensor-level spectrograms across the entire array of gradiometers. See Wiesman and Wilson [44] for a more detailed description of the statistical methodology used to identify the specific time-frequency windows.

The Dynamic Imaging of Coherent Sources (DICS) beamformer was employed to calculate the source power across the entire brain volume [45]. The single images were derived from the cross-spectral densities of all combinations in MEG sensors, and the solution of the forward problem for each location on a grid specified by input voxel space. Following convention, the source power in these images was normalized per subject and trial using an averaged pre-stimulus noise period of equal duration and bandwidth [46,47]. Thus, the normalized power per voxel was computed over the entire brain volume per participant at 4.0 × 4.0 × 4.0 mm resolution, and across trials, this was used to compute pseudo-*t* values per voxel. Each participant’s functional images, which were co-registered to anatomical images prior to beamforming, were transformed into standardized space using the transform previously applied to the structural MRI volume and spatially resampled. The final images (pseudo-*t* units) were used to quantify the relative change in the activity per voxel across the entire cortex. MEG preprocessing and imaging used the BESA software (BESA v6.0; Grafelfing, Germany).

Virtual sensors were extracted from the peak voxels (see below) by applying the sensor weighting matrix derived through the forward computation to the preprocessed signal vector, which resulted in a time series with the same temporal resolution as the original MEG recording [29,48,49]. Once the virtual sensors were extracted, they were transformed into the time-frequency domain and the two orientations for each peak voxel per individual were combined using a vector-summing algorithm. The change in power within each neural region (relative to the baseline) was averaged across the time-frequency window of interest to assess the magnitude of the oscillatory responses seen during the motor planning and execution stages. See Wiesman and Wilson [44] for a more detailed description of our imaging methodology. The Hardy–Weinberg equilibrium test was computed to determine whether the allele frequencies differed from the Hardy-Weinberg equilibrium. Lastly, a Mann–Whitney U-test was utilized to test for differences in the strength of the beta ERD between participants with and without a polymorphism at the *BDNF* gene.

## 3. Results

Twenty-five percent of the participants had a polymorphism at one or both alleles of the *BDNF* gene (*N* = 5; Val66Met or Met66Met = 15.38 ± 1.14 years; GMFCS = I–IV), while seventy-five percent did not have a polymorphism at either of the alleles of the *BDNF* gene (*N* = 15; Val66Val = 16.73 ± 1.25 years, GMFCS = I–IV). This distribution is similar to what has been previously reported for the general population [49] and does not deviate significantly from the Hardy–Weinberg equilibrium (Val frequency = 34 (85%), Met frequency = 6 (15%), Chi-square = 0.93, *p* = 0.335).

The spectrogram permutation tests revealed that there was a significant alpha (10–16 Hz) and beta (18–24 Hz) ERD response (i.e., power decreases) across many sensors near the sensorimotor cortex (*p* < 0.0001, corrected). Given the goals of the study, we focused on the beta response and inspection of the significant time-frequency window indicated that the beta ERD began approximately 250 ms prior to the onset of the isometric force and was sustained while the participants attempted to match the presented force targets (e.g., ~2500 ms; Figure 2). We imaged the beta ERD response from −250 to 250 ms in each participant to capture the movement planning and execution aspects using a baseline period of equal duration and bandwidth (18–24 Hz; −2000 to −1500 ms). The resulting images indicated that the beta ERD was centered on the left precentral gyrus (Figure 3A). The peak voxel within the left precentral gyrus of the grand-averaged image was subsequently used as a seed for extracting virtual sensors (i.e., voxel time series) in each participant. We then calculated the average beta ERD response strength seen across the −250 to 250 ms time window separately in each participant. Our statistical analysis revealed that the individuals with the polymorphism at the *BDNF* gene had a stronger beta ERD than the individuals without a polymorphism (Val66Met and Met66Met = −31.9 ± 5.1%, Val66Val = −18.1 ± 3.3%, U = 61.0, *p* = 0.042, Cohen’s d = 0.63). The reported Cohen’s d indicates a moderate effect of the *BDNF* genotype on the motor-related oscillatory activity.

## 4. Discussion

Our results showed that 25% of the participants had a Val66Met or Met66Met polymorphism at the *BDNF* gene. This percentage is similar to what is reported in the general population [50]. Our results also identified that the sensorimotor cortical beta ERD was stronger in the youth with CP who had the polymorphism at the *BDNF* gene compared to those without it. These results suggest that youth with CP who have a polymorphism at codon 66 of the *BDNF* gene might have greater deficits in the neural circuitry serving planned motor actions [15,51,52,53,54]. Perhaps the stronger beta ERD indicates that those with the polymorphism might activate more pyramidal neurons when attempting to generate a motor action, which may influence the fidelity of the motor action.

The stronger beta ERD seen in those with the polymorphism could alternatively be indicative of aberrant GABAergic inhibitory activity within the sensorimotor cortices. Prior studies have shown that the strength of the beta ERD increases when a GABAA receptor agonist is administered [55], as well as when a GABA transporter is blocked [56]. Thus, increases in GABAergic receptor activity and increased endogenous GABA levels inherently play a role in the generation of the sensorimotor cortical beta oscillations. As such, the increased strength of the beta ERD may be reflective of heightened activity within the underlying GABAergic inhibitory circuits within the sensorimotor cortices of those with the polymorphism. Numerous animal models have demonstrated that disinhibition can result in long-term potentiation, expansion of motor receptive fields, and linking of motor cortical points, suggesting that increased GABAergic inhibition may limit synaptic plasticity [57,58,59]. Thus, increased inhibition in those with the polymorphism may hinder the development of adaptive motor pathways (i.e., reduced neuroplasticity) that could compensate for the original brain insult and reduce the motor planning and execution deficits.

BDNF is a neurotrophin associated with several effects that can impact the entire central nervous system, including neuronal survival, resistance, and modification of receptor sensitivity and synaptic morphology [59,60]. It also catalyzes the synthesis of synaptobrevin and synaptophysin, which are proteins within the SNARE complex that regulate synaptic transmission [60]. Thus, the BDNF protein and its activity-dependent release plays a prominent role in synaptic plasticity [61,62,63,64,65,66,67], and a polymorphism at the *BDNF* gene in neurotypical individuals reduces this activity-dependent release and the capacity for neuroplastic change [21,22]. Potentially, the increased strength of the beta ERD in those with the polymorphism reflects a history of limited synaptic plasticity and thus a reduction in the development of adaptive pathways within the motor system. These cascading effects might play a role in the altered beta ERD seen in those with and without the genetic polymorphism.

The capacity for neuroplastic change is also critical to gaining beneficial outcomes following therapeutic intervention. Currently, there is high variability in therapeutic outcomes in youth with CP, with some individuals displaying substantial sensorimotor improvements and others being clear non-responders. Potentially, the cohort of individuals with CP who have a polymorphism at the *BDNF* gene are more likely to end up in the “non-responder” group. These individuals may benefit from precision or personalized medicine, in which the therapeutic interventions would be more individualized based on *BDNF* genotype. For example, persons with CP that have the *BDNF* polymorphism may require longer or more intense therapeutic protocols to stimulate similar levels of neuroplastic change as their peers without the polymorphism. Alternatively, motor priming utilizing non-invasive brain stimulation may be a viable approach in the near future to alter the membrane excitability of neural populations within the sensorimotor cortices and thereby enhance the capacity for lasting plastic changes [68]. Potentially, this could provide an alternative means for ensuring that the individuals who previously did not respond well to therapy maximize their potential. These are avenues for future work to explore and could enhance our understanding of the variability in treatment outcomes within this population. Of course, to implement this, one would need to test for the polymorphism before designing a therapeutic plan for the individual.

Before closing, it is important to mention some limitations of the current study. First, this study only included persons with spastic CP, and it is unclear how the *BDNF* genotype may affect the motor-related oscillatory activity in persons with other types of CP (e.g., athetoid, ataxia, mixed). Second, we do not have information on whether the participants in the current study have responded well to previous therapeutic interventions. Future work can expand on these findings by uncovering how the polymorphism affects different types of CP, as well as how the *BDNF* genotype may be tied with the variability in therapeutic outcomes across multiple therapies.

## 5. Conclusions

In conclusion, our results suggest that youth with CP who have a single nucleotide polymorphism at the *BDNF* gene might have greater deficiencies in their sensorimotor cortical oscillations while producing a leg motor action. Since prior studies have identified that the *BDNF* polymorphism decreases the capacity for neuroplastic change, it is possible that the altered cortical oscillations are reflective of a reduction in neuroplasticity and thus a decrease in the adaptive pathways that would typically be formed to compensate for the insult to the developing brain. These findings provide a new perspective on the altered beta ERD seen in youth with CP [37,39,41].

## Figures and Tables

**Figure 1 brainsci-12-00435-f001:**
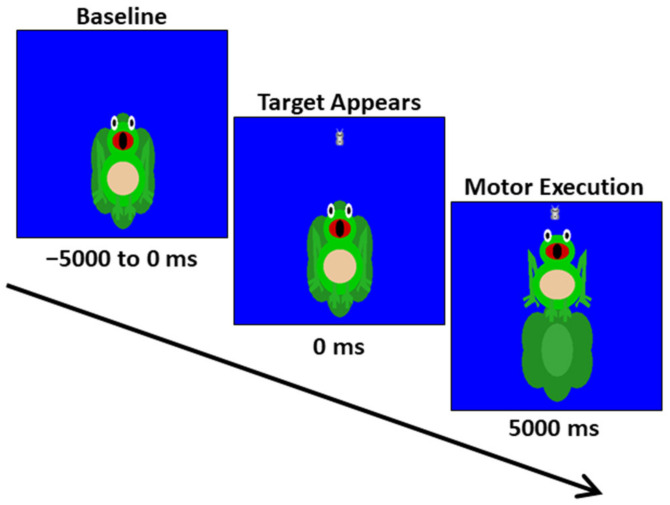
Depiction of the MEG Frog Task experimental paradigm. The trial length was 10,000 ms in total. The participants started each trial at rest for 5000 ms. Subsequently, a target moth would appear, prompting the participant to generate an isometric knee extension force that matched the force value. The target forces varied between 15 and 30% of the participant’s maximum isometric force and were available to be matched for up to 5000 ms. A successful match occurred when the moth that represented the target force was inside of the frog’s mouth for 300 ms. The participants completed 120 target matching trials.

**Figure 2 brainsci-12-00435-f002:**
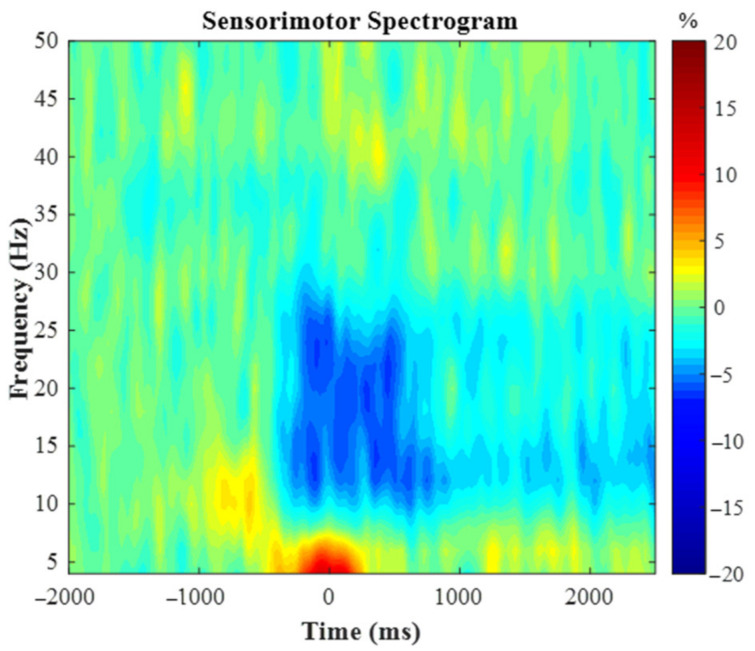
Group-averaged time frequency spectrogram from a sensor located over the leg region of the contralateral (left) motor cortex. Frequency (Hz) is denoted on the y-axis and time (ms) is denoted on the x-axis, with 0 ms representing movement onset of the knee extension. Signal power is expressed as percentage change from baseline with the color bar shown on the right. As discerned, there was a desynchronization within the beta frequency (18–24Hz) that was strongest from −250 to 250 ms.

**Figure 3 brainsci-12-00435-f003:**
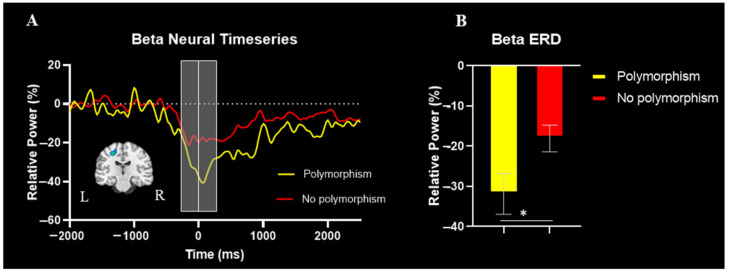
(**A**) Group averaged beamformed image of the beta ERD (18–24Hz, −250 to 250 ms) is shown in the bottom left. The neural time course reflects the response power with percent change from baseline denoted on the y-axis and time denoted on the x-axis. The time interval image is depicted by the light gray box and the white line at time zero reflects movement onset. (**B**) Bar graph representing differences in the strength of the beta ERD between the individuals with CP with and without the polymorphism. The individuals with CP who had a polymorphism at the *BDNF* gene had a stronger beta ERD in comparison to the individuals with CP without a polymorphism (*p* = 0.042). * indicates *p* < 0.05.

## Data Availability

Data are available upon reasonable request to the corresponding author.

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
