# Peer review of "Val66Met Polymorphism Is Associated with Altered Motor-Related Oscillatory Activity in Youth with Cerebral Palsy"

_brainsci, 2022, doi:10.3390/brainsci12040435_

Round 1
Reviewer 1 Report
In their manuscript, Trevarrow et al. investigated a potential link between the Val66Met polymorphism in the BDNF gene and the sensorimotor beta event-related desynchronization in youth with CP. The authors found that the beta ERD was stronger in youth with CP who had the Val66Met or Met66Met polymorphism in comparison to those without the polymorphism.
Overall, the manuscript is written very well and the data are sound. The results are novel and identify BDNF as a potential regulator of sensorimotor cortical oscillations in youth with CP.
I only have two minor suggestions:
In the introduction of the manuscript, the authors refer to two animals studies, that investigated the role of BDNF in the context of motor-skill learning/ sensorimotor cortical reorganization. Please add the study by Andreska et al. (J. Neurosci. 2020) to this paragraph, which puts further weight on the idea of BDNF playing a crucial role in motor skill learning.
In the discussion, the high variability in therapeutic outcomes in youth with CP is mentioned as a potential result of the BDNF polymorphism. Do the authors have any information on the variability in therapeutic outcomes within the investigated group of patients? Although, the individuals have been without treatment for 6 months before conducting the study, is there any information on this before that period? If so, this would nicely add to the discussion.
Author Response
Addressing Reviewer 1’s Comments:
- In their manuscript, Trevarrow et al. investigated a potential link between the Val66Met polymorphism in the BDNFgene and the sensorimotor beta event-related desynchronization in youth with CP. The authors found that the beta ERD was stronger in youth with CP who had the Val66Met or Met66Met polymorphism in comparison to those without the polymorphism. Overall, the manuscript is written very well and the data are sound. The results are novel and identify BDNF as a potential regulator of sensorimotor cortical oscillations in youth with CP.
We thank the reviewer for the positive remarks and for recognizing the importance of our experimental results.
- In the introduction of the manuscript, the authors refer to two animals studies, that investigated the role of BDNF in the context of motor-skill learning/ sensorimotor cortical reorganization. Please add the study by Andreska et al. (J. Neurosci. 2020) to this paragraph, which puts further weight on the idea of BDNF playing a crucial role in motor skill learning.
This citation has been added to the Introduction as suggested.
- In the discussion, the high variability in therapeutic outcomes in youth with CP is mentioned as a potential result of the BDNF polymorphism. Do the authors have any information on the variability in therapeutic outcomes within the investigated group of patients? Although, the individuals have been without treatment for 6 months before conducting the study, is there any information on this before that period? If so, this would nicely add to the discussion.
Unfortunately, we do not have this information. We have added a sentence in the discussion conveying that a better understanding of the link between BDNF genotype and the variability in therapeutic outcomes across multiple therapies is an avenue for future work to explore.
Reviewer 2 Report
This is an exceptionally clear and well written manuscript reporting results of a study investigating whether the strength of movement-related beta ERD response in youth with CP is associated with the BDNF Val66Met polymorphism. The introduction provides a clear background and rationale for the study. Methods appear sound and rigorous. My only suggestion for the methods is that the authors add a brief description of the sample identification and recruitment procedures. The results are clear and complimented by the figures. My only suggestion for the results is that the authors provide the test statistic in addition to the p value and report an effect size estimate and its interpretation. The discussion provides appropriate interpretation and integration of the results with prior literature. I also appreciate the discussion of future clinical implications. The authors might incorporate the “precision medicine” concept into their discussion, as this work falls within this area. The discussion is missing commentary on both limitations of the present study as well as future research directions to improve upon the present study’s limitations and/or continue this line of investigation.
Author Response
Addressing Reviewer 2’s Comments:
- This is an exceptionally clear and well written manuscript reporting results of a study investigating whether the strength of movement-related beta ERD response in youth with CP is associated with the BDNF Val66Met polymorphism.
We thank the reviewer for the positive remarks and for recognizing the importance of our experimental results.
- The introduction provides a clear background and rationale for the study. Methods appear sound and rigorous. My only suggestion for the methods is that the authors add a brief description of the sample identification and recruitment procedures.
The participants were recruited from local clinics and the surrounding community. This information was added to the Methods section as suggested.
- The results are clear and complimented by the figures. My only suggestion for the results is that the authors provide the test statistic in addition to the p value and report an effect size estimate and its interpretation.
As suggested, the effect sizes and test statistics were added to the relevant statistical tests within the Results.
- The discussion provides appropriate interpretation and integration of the results with prior literature. I also appreciate the discussion of future clinical implications. The authors might incorporate the “precision medicine” concept into their discussion, as this work falls within this area.
As suggested, a brief discussion of precision medicine was added to the Discussion.
- The discussion is missing commentary on both limitations of the present study as well as future research directions to improve upon the present study’s limitations and/or continue this line of investigation.
Future directions and limitations were added to the Discussion.
Reviewer 3 Report
In this neurophysiological study conducted on a cohort of 20 patients affected by cerebral palsy, the authors demonstrated that altered sensorimotor beta event-related desynchronization responses were stronger in Val66Met or Met66Met polymorphism carriers.
Although these data could be promising some revisions are needed:
- The introduction is too long, it should be reduced because it is difficult to follow
- Please create short paragraphs to cut the materials and methods section
- Please calculate the Hardy Weimberg equilibrium calculation and single allele frequency in the results
Author Response
Addressing Reviewer 3’s Comments:
- In this neurophysiological study conducted on a cohort of 20 patients affected by cerebral palsy, the authors demonstrated that altered sensorimotor beta event-related desynchronization responses were stronger in Val66Met or Met66Met polymorphism carriers. Although these data could be promising some revisions are needed: The introduction is too long, it should be reduced because it is difficult to follow.
While we respect the reviewer’s suggestion, we have chosen not to substantially modify the introduction. Our reasoning is that each section provides the reader with a comprehensive understanding of the background literature pertaining to the BDNF polymorphism and sensorimotor cortical oscillations. This perspective is also supported by the other Reviewers’ statements that the “introduction provides a clear background and rationale for the study” and that the “manuscript is written very well”.
- Please create short paragraphs to cut the materials and methods section
As suggested, the materials and methods sections have been partially modified to improve flow and for conciseness.
- Please calculate the Hardy Weimberg equilibrium calculation and single allele frequency in the results
The Hardy-Weinberg equation was calculated and indicated that the allele frequency did not differ from the Hardy Weinberg equilibrium (P = 0.335). The single allele frequencies were: Val = 34 (85%), Met = 6 (15%). This information has been added to the results.